# The Implications of the Long Non-Coding RNA *NEAT1* in Non-Cancerous Diseases

**DOI:** 10.3390/ijms20030627

**Published:** 2019-02-01

**Authors:** Felix Prinz, Anita Kapeller, Martin Pichler, Christiane Klec

**Affiliations:** 1Division of Oncology, Department of Internal Medicine, Medical University of Graz, 8036 Graz, Austria; felix.prinz@medunigraz.at (F.P.); anita.kapeller@uni-graz.at (A.K.); martin.pichler@medunigraz.at (M.P.); 2Research Unit of Non-Coding RNAs and Genome Editing in Cancer, Medical University of Graz, 8010 Graz, Austria; 3Department of Experimental Therapeutics, The University of Texas MD Anderson Cancer Center, Houston, TX 77054, USA

**Keywords:** long non-coding RNAs, *NEAT1*, paraspeckles, viral diseases, neurodegeneration

## Abstract

Long non-coding RNAs (lncRNAs) are involved in a variety of biological and cellular processes as well as in physiologic and pathophysiologic events. This review summarizes recent literature about the role of the lncRNA nuclear enriched abundant transcript 1 (*NEAT1*) in non-cancerous diseases with a special focus on viral infections and neurodegenerative diseases. In contrast to its role as competing endogenous RNA (ceRNA) in carcinogenesis, *NEAT1*’s function in non-cancerous diseases predominantly focuses on paraspeckle-mediated effects on gene expression. This involves processes such as nuclear retention of mRNAs or sequestration of paraspeckle proteins from specific promoters, resulting in transcriptional induction or repression of genes involved in regulating the immune system or neurodegenerative processes. *NEAT1* expression is aberrantly—mostly upregulated—in non-cancerous pathological conditions, indicating that it could serve as potential prognostic biomarker. Additional studies are needed to elucidate *NEAT1*’s capability to be a therapeutic target for non-cancerous diseases.

## 1. Introduction

According to the project Encyclopedia of DNA Elements (ENCODE), approximately 70% of the human genome is transcribed into RNA but less than 2% is actually protein-coding [1]. Based on this information, it is not surprising that non-coding RNAs (ncRNAs) have increasingly become a strong focus of research at academic centers [2]. Substantial improvements in sequencing technologies have led to the discovery of the large group of ncRNAs. As indicated in the name, ncRNAs are RNA molecules that are not further translated into proteins. The group of ncRNAs can be divided into subgroups, including the approximately 20-nt long microRNAs (miRNAs), [3,4], Piwi-interacting RNAs (piRNAs) [5], small-interfering RNAs (siRNAs) [6], circular RNAs [7], and long non-coding RNAs (lncRNAs) which are typically >200 nt long [8]. While miRNAs and their significance in cellular regulation were discovered in 1993 [9,10], within the past few years, lncRNAs have emerged as interesting molecules [11,12]. Since then, lncRNAs have been found to be involved in a variety of biological processes, including gene expression regulation, subcellular architecture, and protein complex stabilization [13,14], as well as in physiology and pathophysiology [15,16,17]. Within the last few years, numerous studies have shown that the lncRNA nuclear enriched abundant transcript 1 (*NEAT1*) plays a crucial role in carcinogenesis [18]; emerging evidence, however shows that this lncRNA is also essentially involved in non-cancerous diseases such as neurodegeneration and viral infections. This review aims to provide an overview of information collected thus far pertaining to *NEAT1*’s role in non-cancer related diseases.

## 2. *NEAT1*: Overview, Domain Architecture, Function

The lncRNA *NEAT1* is the main actor in this review and is described in detail, ranging from its discovery and architecture to its cellular and physiological functions.

### 2.1. NEAT1 Overview

Discovered in 2007, *NEAT1* is an un-spliced, polyadenylated non-coding transcript with a high abundance in ovary, prostate, colon, and pancreas [19]. The gene encoding for *NEAT1* is transcribed by Pol II from the multiple endocrine neoplasia locus (*MEN1*) in the human chromosome 11q13 [20]. There are two *NEAT1* isoforms, i.e., a short 3.7 kb (*NEAT1_1*) and a long 23 kb version (*NEAT1_2*) [21]. In contrast to the *NEAT1_2* isoform as observed in high abundance in a subpopulation of cells in the stomach and intestine of adult mice, *NEAT1_1* exhibits a high expression in a wide range of tissues. Accordingly, *NEAT1_2*-dependent paraspeckle formation is primarily detectable in cellular subpopulations of mice [22]. *NEAT1* is—as indicated in the name—enriched in the nucleus and has been shown to function as an essential structural component of paraspeckles and determines the integrity of these subnuclear bodies [19,23]. Cellular depletion of *NEAT1* results in a loss of paraspeckles. Overexpression of *NEAT1*—but not of the paraspeckle component 1 (PSPC1) protein—leads to an increased paraspeckle accumulation pointing towards *NEAT1* being the bottleneck of paraspeckle formation [23]. Paraspeckles are subnuclear ribonucleoprotein bodies composed of the lncRNA *NEAT1* and the core proteins polypyrimidine tract-binding protein PTB-associated splicing factor/splicing factor proline glutamine rich (PSF/SFPQ) [24], 54 kDa nuclear RNA- and DNA-binding protein/non-POU domain-containing octamer-binding protein (p54nrb/NONO) and PSPC1 [25,26]. Paraspeckles are involved in regulating gene expression by a process called nuclear retention. Adenosine-to-inosine (A–I) edited mRNAs are retained in the nucleus whereas unedited mRNAs are transported into the cytoplasm [27,28]. Adenosine-to-inosine editing is a nuclear process which is catalyzed by dsRNA-dependent adenosine deaminases (ADARs) leading to a hydrolytic deamination of adenosine to inosine in double-stranded regions of targeted mRNAs. Adenosine-to-inosine editing predominantly occurs in inverted repeated *Alu* elements (IR*Alu*s) [28,29]. Paraspeckles can therefore be seen as nuclear mRNA anchors [30].

### 2.2. NEAT1 Domain Architecture

Two of the paraspeckle protein components (p54nrb/NONO and PSPC1) were found to form heterodimers within paraspeckles [31] and exhibit an intensive co-localization with *NEAT1* [23]. Three protein interaction sites located near the 5’ and 3’ ends of *NEAT1* are a prerequisite in order to bind to p54nrb/NONO [32] (Figure 1A). An advanced combination of protein and RNA visualization techniques, namely structured illumination microscopy (SIM) and fluorescent in situ hybridization (FISH), allowed for the simultaneous detection of *NEAT1* and the protein components within paraspeckles [33]. *NEAT1* has been shown to arrange itself in a core-shell spheroidal structure. The 5’ and 3’ ends of *NEAT1* are located at the periphery of the speckles whereas the central sequence is localized within the core [34] (Figure 1B). 

Both *NEAT1*’s middle domain and the binding of p54nrb/NONO to this central site are two features essential and sufficient for paraspeckle formation [35]. Li et al. [36] provided data showing that only the long 23 kb *NEAT1* isoform is a major and essential component of paraspeckles, suggesting that the short isoform might be implicated in other cellular functions. Lin et al. [37] provided a structural model of the long *NEAT1* isoform showing that long-range interactions between its 5’ and 3’ ends may be important for its architectural role within paraspeckles. Several studies demonstrate that the structure of lncRNAs is defining their function. LncRNAs tend to acquire complex secondary and tertiary structures and it was observed that structural conservation rather than nucleotide sequence conservation is crucial for maintaining their function [38]. In addition to its domains relevant for paraspeckle assembly, *NEAT1*’s domain architecture and interaction with other proteins is also regulating global pri-miRNA processing. As described above, *NEAT1* interacts with p54nrb/NONO and PSF/SFPQ as well as with other RNA-binding proteins. Furthermore, it possesses multiple RNA segments including a “pseudo pri-miRNA” near the 3’ end, which—together with the above mentioned protein-*NEAT1* interactions—help to attract the microprocessor Drosha/DiGeorge syndrome critical region 8 (DGCR8) complex responsible for pri-miRNA processing [39].

### 2.3. Cellular Function of NEAT1

In the cellular context, *NEAT1_2* is responsible for the sequestration of paraspeckle components in the drosophila behavior human splicing (DBHS) family, i.e., PSF/SFPQ, PSPC1, and p54nrb/NONO. PSPC1 and p54nrb/NONO regulate the A–I editing of mRNAs [23] and additionally, p54nrb/NONO is involved in retaining those edited mRNAs, preventing their nuclear export [40]. *NEAT1* itself is retained in the nucleus without being A–I edited [23]. Depending on the presence of *NEAT1* and the associated abundance of paraspeckles in the nucleus, gene expression regulation is influenced to a variable extent. In the case of low paraspeckle accumulation within the nucleus, the high concentration of unbound paraspeckle protein components PSPC1, PSF/SFPQ, and p54nrb/NONO, results in an increased transcriptional regulation of specific genes through their action as positive or negative transcription regulators. Furthermore, a low amount of paraspeckles leads to a decreased nuclear retention of A–I edited mRNAs, thus leaving the export of those mRNAs to the cytoplasm uninfluenced. Conversely, increased paraspeckle formation means a lower concentration of unbound paraspeckle components, therefore limiting their effect on the transcriptional regulation. In addition, A–I edited mRNAs are more efficiently bound by paraspeckles and in turn more effectively retained in the nucleus instead of being transported to the cytoplasm [41] (Figure 2). 

Another mode of action of *NEAT1* is sponging of miRNAs, which is a function already well documented in carcinogenesis [18] and described in Section 2.5.

### 2.4. NEAT1 in Physiology

Nakagawa et al. [22] propose that paraspeckles are non-essential, cell subpopulation specific nuclear bodies, since *NEAT1* knock-out mice are viable, fertile, and do not show an apparent phenotype. In a subsequent study, the same group provided data that *NEAT1_2*-mediated paraspeckles are fundamental for corpus luteum formation and contradictory to the earlier results partially define fertility in a subpopulation of mice [42]. Furthermore, *NEAT1* was shown to be essential for mammary gland development and the lactation capacity in mice [43] as well as modulating neuronal excitability in humans [44]. As Chen and Carmichael [28] demonstrated, *NEAT1_2* is also involved in the differentiation of human embryonic stem cells (hESC). Although hESC mRNAs contain IR*Alu* elements, and thus, are likely to be A–I edited, these mRNAs are not retained in the nucleus but are transported into the cytoplasm. They are subjected to nuclear retention only after differentiation and the reason why can be found when looking at the difference between *NEAT1_2* expression levels. While undifferentiated hESC lack *NEAT1_2*-dependent paraspeckles (which, in turn, regulate nuclear retention), *NEAT1_2* expression is initiated by differentiating enabling the paraspeckle formation and subsequent nuclear retention of hESC mRNAs [28]. *NEAT1_2* serves an additional functional role in the transcriptional regulation of Interleukin-8 (IL-8). In a viral infection, *NEAT1* sequesters the IL-8 repressor PSF/SFPQ from the IL-8 promoter to paraspeckles. As a result, IL-8 transcription is initiated, leading to immune response stimulation [41].

### 2.5. NEAT1 in Carcinogenesis

Meta-analyses demonstrate an intensive upregulation of *NEAT1* in several cancer entities, resulting in an unfavorable outcome as well as a drastic decrease in overall survival, suggesting a potential role for *NEAT1* as prognostic biomarker [45,46]. Numerous studies focusing on *NEAT1*’s role in cancer biology indicate that this lncRNA is a crucial part of carcinogenesis as found in non-small lung cancer [47,48,49,50], breast cancer [51,52,53], hepatocellular carcinoma [54,55,56,57], ovarian cancer [58,59,60,61], and prostate cancer [62,63], just to name a few. In terms of carcinogenesis, *NEAT1* mainly functions as competing endogenous RNA (ceRNA) by sponging tumor-suppressive miRNAs [64]. Subsequently, these miRNAs lose the ability to function as a tumor suppressor and oncogenic mRNAs are translated, ultimately contributing to tumorigenesis [65]. As this review focuses on *NEAT1* in non-cancerous diseases, we will refrain from providing more details. For a recent overview of *NEAT1*’s role in carcinogenesis, please see Klec et al. [18].

## 3. Immune System and Viral Diseases

Over the past ten years, miRNAs and lncRNAs have become key players in immune system response as well as cancer immunotherapy [66,67]. LncRNAs have been demonstrated to be expressed in a lineage-specific manner, for example, in T-cell population subsets [68] or in a certain subset of lymphocytes [69]. Furthermore, lncRNAs were shown to be involved in controlling the differentiation and function of innate and adaptive immune cell types. Examples of lncRNAs in immune system regulation include: (1) the lncRNA *H19* regulates hematopoietic development [70], (2) *Morrbid* and *lnc-DC* are involved in the regulation of myeloid cell survival and myeloid cell differentiation [71,72], (3) *lincR-Ccr2-5’AS* controls CD4+ T-cell differentiation [73], and (4). *lincRNA-Cox2* is an activator of inflammation [74]. Chen et al. [75] provided a detailed overview of lncRNAs in immune system regulation. 

Concerning *NEAT1*, its involvement in immune system responses was initially discovered in the brains of mice infected with either Japanese encephalitis virus or the rabies virus [13,76]. Since this discovery, *NEAT1*’s role in regulating the immune system has been studied intensively and is summarized in the next section and in Table 1.

### 3.1. Sepsis and Sepsis-Induced Acute Kidney Injury

Non-coding RNAs have been shown to play an important role in the host defense system and viral infections [77,78]. *NEAT1* was shown to be differentially expressed and is proposed to be a suitable candidate as an additive biomarker for early sepsis detection as it was found to be highly upregulated in peripheral blood mononuclear cells (PBMCs) in sepsis patients compared to healthy controls [79], and in sepsis-induced acute kidney injury (AKI) [80]. In the case of sepsis-induced AKI, *NEAT1* expression correlates positively with the severity of the disease. Knock-down of *NEAT1* in rat kidney cells leads to reduced lipopolysaccharide (LPS)-induced cell injury by an accompanied upregulation of *miR-204. miR-204* has been shown to protect the kidney during sepsis by regulating Hmx1 (heme oxygenase) [81]. These protective functions are lost when increased *NEAT1* levels sponge *miR-204*, activating NF-κB signaling and leading to sepsis associated organ failure [82] in sepsis-induced AKI patients [80].

### 3.2. Viral-Induced Diseases

As a stress-induced lncRNA, *NEAT1* expression increases in response to a viral infection. Over the last two years, numerous viral diseases have been correlated with differences in *NEAT1* expression, either acting anti-viral or pro-viral. 

#### 3.2.1. Anti-viral Effects of *NEAT1*

*NEAT1* was first linked to an infection with the human immunodeficiency virus (HIV-1) in 2013. Zhang’s group [83] detected an increased *NEAT1* expression after HIV-1 infection. A new finding at that time, the paraspeckle components PSF/SFPQ, p54nrb/NONO, and matrin 3 had until then only been associated with HIV infection. Knockdown of *NEAT1* in the T-cell lines Jurkat and MT4 resulted in an increased HIV-1 production due to a more pronounced nucleus-to-cytoplasm export of Rev-dependent instability elements (INS)-containing *HIV-1* mRNA, or in other words, decreased paraspeckle-mediated nuclear retention [83]. A subsequent study could shed more light on how the underlying mechanism is regulated. Exportin 1 (XPO1) is the Rev-dependent mediator of the abovementioned nuclear export. The nucleus-to-cytoplasm transfer of these incompletely spliced viral transcripts regulates gene expression post-transcriptionally. The export of these un-spliced transcripts is inhibited due to inefficient splicing and the contained INS elements. As paraspeckles depend crucially on *NEAT1* and have been shown to retain these INS-containing transcripts in the nucleus, it was proposed that increased HIV-1 replication after *NEAT1* knock-down is a consequence of decreased nuclear retention of *HIV-1* mRNA [84]. Pandey et al. [85] reports on differential *NEAT1* expression in dengue disease, proposing that *NEAT1* could be a marker for the progression of dengue fever since its expression is reduced in severe phenotypes found in dengue disease. Interestingly, *NEAT1* expression was highest in the early stages of the disease, decreasing with progression [85]. The authors suggest that low *NEAT1* expression could induce apoptosis via p53 [86] in monocytes, as already reported in myeloid lineage cells [76] or breast cancer cell lines [51]. Hantavirus infection is followed by increased *NEAT1* expression which, on the one hand, controls viral replication and induces anti-viral immune response by promoting interferon (IFN) production via retinoic acid inducible gene I (RIG-I) signaling, on the other hand. The mechanism behind is the above described SFPQ sequestration to paraspeckles away from the RIG-I promoter which is known to be a positive regulator of IFN-gene activation [87], therefore, promoting anti-viral immunity [88,89]. Beeharry et al. [90] demonstrated that the Hepatitis D virus (HDV) interacts with the major paraspeckle components, i.e., PSF/SFPQ, PSPC1, and p54nrb/NONO, indicating a crucial role of paraspeckles in HDV infection. Indeed, viral replication was shown to depend crucially on this interaction as a knockdown of these proteins leads to a hampered HDV replication. Upon HDV infection, *NEAT1* levels are upregulated and *NEAT1* foci are enlarged. Due to the facts that IL-8 levels are 2-fold increased upon HDV infection and a knockdown of paraspeckle proteins results in reduced HDV replication, it is tempting to speculate that *NEAT1* upregulation-induced sequestration of paraspeckle proteins is causing the anti-viral effects leading to the activation of innate immunity [90,91]. The upregulation of IL-8 is based on the *NEAT1*-induced relocation of the paraspeckle component SFPQ to paraspeckles where it is unable to execute its repressor function on IL-8 transcription [91].

#### 3.2.2. Pro-viral Effects of *NEAT1*

Recently, two independent groups provided data on *NEAT1* upregulation after an infection with the herpes simplex virus (HSV) [92,93]. Viollet et al. [92] demonstrated that approximately 210 genes are upregulated after Kaposi sarcoma-associated herpesvirus (KSHV) infection, with *NEAT1* showing a 3-fold increase in expression in KSHV infected cells versus non-infected cells under hypoxic conditions. The hypoxia-inducible factor 2 (HIF2) is a known regulator of *NEAT1* transcription [94]. Since *NEAT1* has been demonstrated to increase survival of cancer cells [18,94], the authors propose that an upregulation of HIF-responsive genes causes tumorigenesis leading to the development of Kaposi sarcoma and other KSHV-induced tumors [92]. Wang et al. [93] showed that *NEAT1* is upregulated after a herpes simplex infection in a STAT3-dependent manner. The HSV-1 genome is recruited to paraspeckles to regulate its transcription. *NEAT1* upregulation facilitates virus replication by mediating the interaction between the paraspeckle components p54nrb/NONO, and PSPC1 and herpes simplex gene promoters, ultimately leading to an increased viral gene expression. Knockdown of the anti-viral SFPQ results in increased replication due to facilitation of the interaction between STAT3 and viral gene promoters. Initial advances pertaining to therapeutic interventions by *NEAT1* modulation show that in the case of herpes simplex infection, thermosensitive gels coated with siRNA against *NEAT1* were able to reduce virus induced skin lesions [93]. These data show on the one hand that HSV-1 replication is regulated by a *NEAT1*-dependent paraspeckle-mediated transcriptional cascade and on the other hand that *NEAT1* upregulation seems to be a general response to viral infections. If the consequences are pro-viral or anti-viral depends on the downstream mechanisms.

## 4. Neurodegeneration and Neuronal Defects

Numerous studies show that lncRNAs are involved in regulating and/or protecting against neurodegeneration [95]. There are several indications underlining the importance of lncRNAs in brain function and the development of neurodegenerative diseases: (1) the existence of brain-specific lncRNAs with precisely regulated temporal and spatial expression patterns [96]; (2) the correlation between highly transcriptional active CNS cells and the fact that lncRNAs are involved in transcriptional regulation; (3) the tissue-specific expression of certain lncRNAs either in particular regions of the CNS or even in different cell types [97,98]; and (4) the observation that dysregulations or mutations in lncRNA gene loci are associated with neurodegenerative disorders [99]. Several lncRNAs have been demonstrated to be involved in the development and progression of neurodegenerative diseases. Some well-studied examples include the following: (1) the β-site amyloid β-protein precursor (APP) cleaving enzyme 1 antisense RNA (*BACE1-AS*) in Alzheimer’s disease (AD); (2) phosphatase and tensin homolog (PTEN)-induced kinase 1 antisense (*PINK1-AS*) which is stabilizing its protein-coding pendant PINK1 in Parkinson’s disease (PD) [100,101,102]; and (3) brain-derived neurotrophic factor antisense (*BDNF-AS*) in Huntington’s disease (HD) [103,104]. More detailed information about the role of lncRNAs in neurodegenerative diseases are well reviewed by Quan et al. [99] and Wan et al. [105].

*NEAT1* is speculated to have biological functions in the brain’s pathologies since the expression level of this lncRNA increases significantly in the nucleus accumbens of heroin users [106] and is described as a mediator of the neuroprotective effects of bexarotene on traumatic brain injury in mice [107]. Table 2 summarizes the neurodegenerative diseases with *NEAT1* contribution, which are also discussed in detail below.

### 4.1. Huntington’s Disease

HD is caused by an expansion of a CAG triplet repeat within the *huntingtin* gene, creating a mutant version of the huntingtin protein [108]. There is some controversy as to whether or not *NEAT1* actively contributes to Huntington’s disease pathogenesis or if it triggers neuroprotective mechanisms. Microarray analyses as well as quantitative PCR of post-mortem brains of Huntington’s disease patients and R6/2 mouse brains—a model for HD—show an upregulation of *NEAT1* expression. Overexpression of *NEAT1* in neuro2A cells increases cell viability upon neuronal injury induced by H_2_O_2_ treatment, thus the authors suggest a protective mechanism against oxidative injury in HD pathogenesis rather than a contribution to disease development [109]. A review by Johnson et al. [110] reports contrary findings. Chip sequencing indicates that *NEAT1* is a target of REST (a transcriptional repressor) and p53 (a tumor suppressor), both of which are known to be key players in HD. Changes in lncRNA expression are considered to result in altered epigenetic gene regulation in diseased neurons, and thus are believed to be possible contributors to HD pathology [110]. Further studies are needed to unravel *NEAT1*’s role in HD pathogenesis.

### 4.2. Multiple Sclerosis

Multiple sclerosis (MS) is a chronic autoimmune, inflammatory neurological disease of the central nervous system (CNS) [111]. A screening of 84 lncRNAs revealed an upregulation of *NEAT1* in the serum of patients suffering from multiple sclerosis (MS) compared to either healthy controls or patients with who have idiopathic inflammatory myopathy (IIM) [112]. Taking into consideration that *NEAT1* plays a role in immunity by driving IL-8 activation (in addition to findings by Lund et al. [113] which demonstrate that IL-8 levels increase significantly in MS patients), the authors speculate that upregulated *NEAT1* expression activates IL-8 transcription. In addition to IL-8 upregulation, a co-localization of stathmin and toll-like receptor 3 (TLR3) has been found in astrocytes, microglia, and neurons in the brains of MS patients [114], indicating that these players contribute to MS pathogenesis [112]. 

### 4.3. Amyotrophic Lateral Sclerosis

Amyotrophic lateral sclerosis (ALS) is a fatal motor neuron disorder which is characterized by progressive loss of the upper and lower motor neurons [115]. *NEAT1* is upregulated in ALS patients in the early stages of the disease. Nishimoto et al. [116] were able to show a direct interaction of *NEAT1* with two RNA-binding proteins, both of which are mutated in ALS patients, i.e., TAR DNA-binding protein 43 (TDP-43) [117] and fused in sarcoma/translocated in liposarcoma (FUS/TLS) [118]. Both proteins were shown to be enriched in paraspeckles of cultured cells. Interestingly, *NEAT1* expression was absent in motor neurons in the spinal cords of healthy control mice but there was a high density of *NEAT1* as well as paraspeckles in motor neurons in the spinal cords of ALS patients in the early phase of disease. Based on these results, the authors claim that paraspeckles directly contribute to neurodegenerative diseases [116].

### 4.4. Parkinson’s Disease

Parkinson’s disease (PD) is a chronic, progressive movement disorder due to a loss of dopamine producing cells in the brain [119]. *NEAT1* is upregulated in a 1-methyl-4-phenyl-1,2,3,6-tetrahydropyridine (MPTP)-induced PD mouse model together with PINK1 (a known contributor to PD [120]) which was stabilized by *NEAT1*. Since a knockdown of *NEAT1* leads to reduced MPTP-initiated autophagy in vivo resulting in a decreased neuronal injury, authors claim that there is a contribution of *NEAT1* in PD. [121] Nearly simultaneously, Liu’s group [122] published supportive results by also showing an upregulation of *NEAT1* in an MPTP-induced PD mouse model as well as in a 1-methyl-4-phenylpyridinium (MPP+)-induced PD cell line. *NEAT1* knockdown resulted in an increased viability, inhibition of apoptosis, and decreased α-synuclein expression in the PD cell model. Since α-synuclein overexpression reversed the effects of *NEAT1* knockdown, a protective role of *NEAT1* downregulation in the MPTP-induced PD mouse model was suggested [122].

## 5. Conclusions

Summarizing recent data pertaining to *NEAT1*, one can draw the conclusion that this lncRNA contributes to regulating viral diseases and neurodegeneration. One shared feature of the so far investigated non-cancerous diseases is a common upregulation of *NEAT1* which highlights the enormous potential of this lncRNA as diagnostic biomarker in these pathologies. Although increased *NEAT1* levels seem to be a common event upon viral infection and in neurodegenerative diseases, the consequences of *NEAT1* upregulation are diverse. We hypothesize that the mode of *NEAT1*’s action—if it acts pro-viral or anti-viral or if it is contributing to neurodegeneration or protecting from it—depends on the occurring downstream events. From the mechanistic point of view, *NEAT1* either exerts the function of paraspeckle-mediated nuclear retention (as in HIV infection), transcriptional regulation by sequestering paraspeckle proteins (as in HTNV, HDV and Herpes simplex infections as well as in Multiple Sclerosis) or sponging of miRNAs (as in sepsis-induced AKI). Concerning the other discussed diseases more studies are needed to investigate the consequences of *NEAT1* upregulation and the underlying mechanisms. Although nuclear retention and transcriptional regulation by sequestration of paraspeckle proteins seem to be *NEAT1*’s predominant modes of action in non-cancerous diseases, there is emerging evidence that it also can act as sponge for miRNAs. For instance, Wang et al. [123] observed that, in the context of diabetic nephropathy (DN) progression, *NEAT1* directly binds to *miR-27b-3p*, leading to the suppression of its function. As *miR-27b-3p* directly targets *ZEB1* (a key player in the EMT process), the group suggests that inhibition of *NEAT1* represses DN progression through regulating EMT (and fibrogenesis). Further evidence which consolidates *NEAT1*’s ability to sponge miRNAs was provided by Wang et al. [124] by demonstrating that *NEAT1* directly binds to *miR-342-3p* in the context of atherosclerotic cardiovascular diseases. Wang’s group observed that a knockdown of *NEAT1* represses the inflammation response and inhibits lipid uptake by THP-1 cells in a *miR-342-3p*-dependent manner. Chen et al. [125] corroborate *NEAT1*’s ability to sponge miRNAs in the context of atherosclerosis. Their group observed that *NEAT1* directly binds to *miR-128* and through that interaction plays a role in oxidized low density lipoprotein (ox-LDL)-induced inflammation and oxidative stress in atherosclerosis development. Despite the fact that there are reports on *NEAT1* influencing disease progression by sponging miRNAs, the described paraspeckle-mediated effects on the transcriptional regulation seem to be of greater significance in the context of non-cancerous diseases. *NEAT1*’s suitability as therapeutic target still needs more intensive research. As already mentioned above, first advances towards a therapeutic application of *NEAT1* have been made in the context of Herpes simplex infection where virus-induced skin lesions have successfully been treated with gels containing *NEAT1* siRNA [93]. Therefore, we believe that *NEAT1* plays a crucial role in non-cancerous diseases and future studies will help to understand the complete story of this interesting lncRNA.

## Figures and Tables

**Figure 1 ijms-20-00627-f001:**
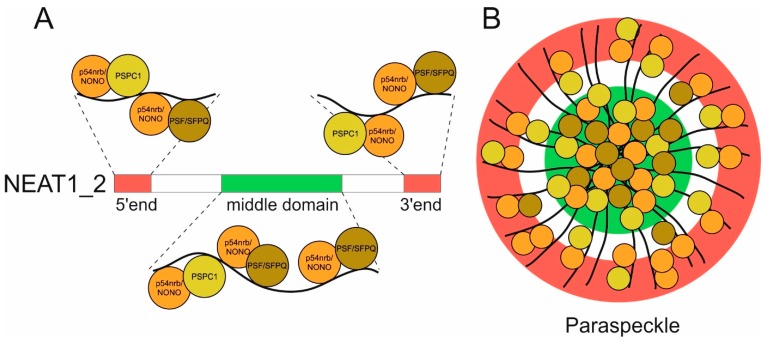
Nuclear enriched abundant transcript 1 (*NEAT1*)’s domain architecture and schematic paraspeckle structure. (**A**) The long isoform of *NEAT1* (*NEAT1_2*) contains three domains which are relevant for binding the paraspeckle-associated proteins 54 kDa nuclear RNA- and DNA-binding protein/non-POU domain-containing octamer-binding protein (p54nrb/NONO; orange), paraspeckle component 1 (PSPC1; yellow), and polypyrimidine tract-binding protein PTB-associated splicing factor/splicing factor proline glutamine rich (PSF/SFPQ; brown). p54nrb/NONO and PSF/SFPQ directly interact with *NEAT1*’s middle domain, whereas three protein interaction sites near the 5’ and the 3’ end facilitate binding of p54nrb/NONO. All three abovementioned proteins form heterodimers in every possible combination, and thus contribute to the formation of paraspeckles. Only proven interactions of proteins with *NEAT1* and each other are shown. (**B**) Paraspeckles are arranged in a spheroidal, highly ordered structure with *NEAT1*’s middle domain being located in the center while its 5’ and 3’ termini are at the periphery of the structure. Paraspeckle-associated proteins p54nrb/NONO, PSF/SFPQ, and PSPC1 (as well as other paraspeckle proteins; not shown in figure) are distributed within the structure in consideration of the before established binding domains on *NEAT1*.

**Figure 2 ijms-20-00627-f002:**
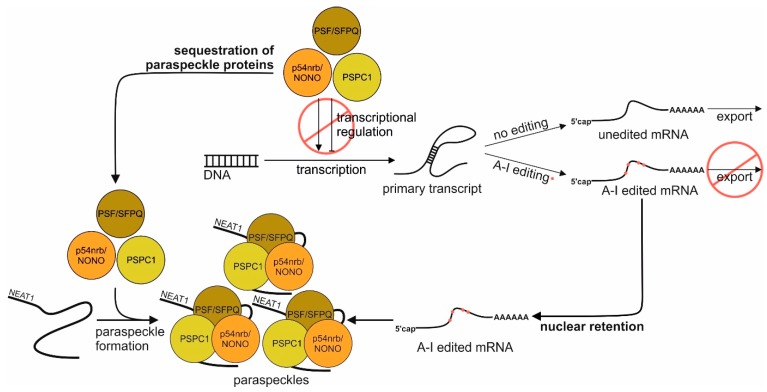
Role of *NEAT1* in the regulation of gene expression. *NEAT1* influences gene regulation through two predominant functions. On one hand, *NEAT1*-dependent paraspeckle formation leads to the sequestration of paraspeckle proteins such as PSPC1, PSF/SFPQ, and p54nrb/NONO, therefore limiting their effect on the transcriptional regulation. On the other hand, adenosine-to-inosine (A–I) edited mRNAs are more efficiently bound by formed paraspeckles and in turn more effectively retained in the nucleus, instead of being transported to the cytoplasm.

**Table 1 ijms-20-00627-t001:** *NEAT1* expression levels in several viral-induced diseases together with the proposed molecular pathway.

*NEAT1* Expression	Viral Disease	Pro-viral or Anti-viral	Pathway	Literature
Upregulation	HIV-1	anti-viral	Rev-dependent nuclear export	[83,84]
	Mild dengue			[85]
	Herpes simplex	pro-viral	P54nrb, PSPC1	[91,93]
	Hantavirus	anti-viral	RIG-I-signaling	[88,89]
	Hepatitis D	anti-viral	IL-8 induction	[90]
	Influenza	anti-viral	IL-8 induction by SFPQ inhibition	[91]
Down-regulation	Severe dengue	anti-viral	p53 induced apoptosis	[85]

**Table 2 ijms-20-00627-t002:** *NEAT1* expression levels in several neurodegenerative diseases and predicted co-players.

*NEAT1* Expression	Neurodegenerative Disease	Co-players	Literature
Upregulation	Huntington’s Disease	REST, p53	[109,110]
	Multiple Sclerosis	IL-8, stathmin & TLR3	[112,113,114]
	Amyotrophic Lateral Sclerosis	TDP-43, FUS/TLS	[116]
	Parkinson’s Disease	α-synuclein	[121,122]

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
