# Peer review of "The Implications of the Long Non-Coding RNA NEAT1 in Non-Cancerous Diseases"

_ijms, 2019, doi:10.3390/ijms20030627_

Reviewer 1 Report

Review of “Implications of the long non-coding RNA NEAT1 in non-cancerous diseases” by Prinz et al., 2018

In this review on NEAT1’s role in non-cancerous diseases, the authors attempt to bring clarity to the wealth of data about NEAT1 as well as point to areas where data are lacking. The review is overall quite informative and well cited. I feel that some reorganization, condensing a few areas, and some moderate grammatical revision will make this manuscript suitable for publication. As this journal is directed towards an English speaking audience, I would have it edited one more time by someone for whom English is a primary language to resolve some of the more nuanced peculiarities of the English language. Below are a few suggestions that would make the manuscript tighter and less redundant.

1.       In section 2.2, labeled NEAT1 structure, I felt like this would be more appropriately labeled NEAT1 domain architecture, as the review does not discuss the actual 3D shape of NEAT1 at all, but instead domains that are necessary for function. In addition, I would add the information about the two isoforms (lines 78-81) to section 2.2 as it partially redundant between the two section.

2.       In section 2.3 (NEAT1 function), are the functions described in lines 84-100 a result of the long or short isoform?

a.       Additionally, are these direct, indirect, or correlative “functions”? Also, some of these “functions” are merely phenotypes from loss (or miss-regulation) of NEAT1.

b.       How can NEAT1 be both non-essential and fundamental for fertility in mice (lines 82 and 84). This may go back to the long and short isoform-dependent roles of NEAT1.

c.       Finally, there is a mixture of NEAT1 localization and function in section 2.3 that perhaps should not be mixed together under that heading? To me, NEAT1 function would be its biochemical functions that result from direct interactions with other proteins, RNA, or genomic loci, not its localization or things that correlate with its expression. In addition, it seems that some of this information appears later on and therefore is redundant.

3.       On line 136, what does LPS mean?

4.       Section 3.2 is long and does not really bring clarity to the role of NEAT1 in viral infection. I would suggest breaking this section up into two sub sections that discuss ways in which NEAT1 in pro infection and another in which it inhibits infection.

a.       Is upregulation of NEAT1 a general response to viral infection (referring to line 142) and the differing outcomes are all a result of things that occur downstream of NEAT1 upregulation?

5.       Section 4 changes tenses back and forth between past and present. In addition, this introductory section discusses many of the topics that are then discussed again in the sub sections (4.1-4.4). I would suggest cutting out much of this introductory section as moving non-redundant information to the below sub sections.

6.       The conclusion section is primarily repeating what is found in the previous section and doesn’t really summarize the information so well. This section could be shortened to present a more “general” view of what NEAT1 is doing (leaving it at just lines 259-265 and removing lines 277-291).

a.       If the information found in lines 277-291 are retained, please change the phrase “more data is” to “more data are” (line 290).

To summarize, it appears that NEAT1 has one main function – sequestration of proteins and mRNAs from their own intended targets or molecular outcomes. This was not really clear until the conclusion (perhaps because of the way in which the function section was organized). All pathologies associated with NEAT1 must be described and understood in this light. I think if this single point is made more clear, and distracting information is removed, then this manuscript will make a much larger contribution to our synthesis and understanding of the role of NEAT1 in disease.

Author Response

Dear Reviewer 1

The manuscript was substantially changed and restructured according to your valuable comments and suggestions. We highly appreciate the detailed reviews which helped us improving the quality of the manuscript immensely. We tried to address every point as detailed as possible and list the responses to your comments in point-by-point fashion below.

As this journal is directed towards an English speaking audience, I would have it edited one more time by someone for whom English is a primary language to resolve some of the more nuanced peculiarities of the English language.

The manuscript was proofread and edited by native-speaker Nadine Lichtenberger, which drastically improved the style and language of the manuscript.

1. In section 2.2, labeled NEAT1 structure, I felt like this would be more appropriately labeled NEAT1 domain architecture, as the review does not discuss the actual 3D shape of NEAT1 at all, but instead domains that are necessary for function. In addition, I would add the information about the two isoforms (lines 78-81) to section 2.2 as it partially redundant between the two section.

We have changed the subheading of 2.2 into “NEAT1 Domain Architecture” and now the subheading is very well representing the content of the paragraph. Additionally, the information about the two isoforms has been shifted to section 2.1 (lines 49-53).

2. In section 2.3 (NEAT1 function), are the functions described in lines 84-100 a result of the long or short isoform?

We clarified in the text that these functions go back to the long isoform (i.e. NEAT1_2).

a. Additionally, are these direct, indirect, or correlative “functions”. Also, some of these “functions” are merely phenotypes from loss (or miss-regulation) of NEAT1.

We have totally restructured the section about NEAT1 function and think it gives a better overview of NEAT1’s functions.

b. How can NEAT1 be both non-essential and fundamental for fertility in mice (lines 82 and 84). This may go back to the long and short isoform-dependent roles of NEAT1.

We tried to clarify that NEAT1 is non-essential but is needed for a full fertility-capacity in a subpopulation of mice. Half of the NEAT1 KO mice show a reduced fertility. This inconsistency goes back that the same group in the first publication mentioned that mice are fertile and three years later provide data that NEAT1 defines the fertility in a subpopulation of mice.

c. Finally, there is a mixture of NEAT1 localization and function in section 2.3 that perhaps should not be mixed together under that heading? To me, NEAT1 function would be its biochemical functions that result from direct interactions with other proteins, RNA, or genomic loci, not its localization or things that correlate with its expression. In addition, it seems that some of this information appears later on and therefore is redundant.

We have shifted the expression of the two isoforms to section 2.1. Additionally, we have divided the section 2.3. “NEAT1 function” into two sections: 2.3. “Cellular Function of NEAT1” (describing nuclear retention of mRNAs, transcriptional regulation via sequestration of paraspeckle proteins, as well as miRNA-sponging) and 2.4. “NEAT1 in Physiology” (describing the so far investigated effects of NEAT1 on physiologic events such as lactation capacity, neuronal excitability, corpus luteum formation, hESC differentiation and viral infection).

3. On line 136, what does LPS mean?

LPS stands for lipopolysaccharide – the full name is now mentioned in the text as well as in the abbreviations list.

4. Section 3.2 is long and does not really bring clarity to the role of NEAT1 in viral infections. I would suggest breaking this section up into two sub sections that discuss ways in which NEAT1 is pro infection and another in which it inhibits infection.

Due to the valuable comment of the reviewer, we divided section 3.2 into two parts (3.2.1. Anti-viral Effects of NEAT1 and 3.2.2. Pro-viral Effects of NEAT1) and now the text is not as dense as before and much more reader-friendly.

a. Is upregulation of NEAT1 a general response to viral infection (referring to line 142) and the differing outcomes are all a result of things that occur downstream of NEAT1

This point has been addressed in the Conclusion section (Lines 317-319).

5. Section 4 changes tenses back and forth between past and present. In addition, this introductory section discusses many of the topics that are then discussed again in the sub sections (4.1-4.4). I would suggest cutting out much of this introductory section as moving non-redundant information to the below sub sections.

The whole manuscript was proofread and edited by a native-speaker, therefore, no more tense-switching is present in the manuscript. A huge part of the mentioned introductory section was removed and only three short examples of lncRNAs involved in neurodegeneration are mentioned to highlight the importance of lncRNAs in neurodegeneration. A short introductory sentence on the diseases has been added at the beginning of each subsection (4.1.-4.4.).

6. The conclusion section is primarily repeating what is found in the previous section and doesn’t really summarize the information so well. This section could be shortened to present a more “general” view of what NEAT1 is doing (leaving it at just lines 259-265 and removing lines 277-291).

We have removed the suggested part and re-structured the conclusion. It now gives a more general view of what NEAT1 is doing and how it acts in non-cancerous diseases (i.e. the common upregulation of NEAT1, nuclear-retention of mRNAs, transcriptional regulation due to sequestration of paraspeckle proteins or sponging of miRNAs).

a. If the information found in lines 277-291 are retained, please change the phrase “more data is” to “more data are” (line 290).

The respective section was removed.

Thank you very much for your efforts.

Sincerely,

Felix Prinz & Christiane Klec

Reviewer 2 Report

This is a very thorough review of the current knowledge about NEAT1 functions in non-cancerous diseases by Prinz et al. As lncRNA are highly abundant and many studies have shown them to be required for efficient and productive viral replication and involvement in dysregulation of myriad of cellular processes. The structure-to-function relationship of lncRNAs will continue to be important as these lncRNAs are beginning to be looked at as targets for antiviral treatments.

This reviewer has couple suggestions when it comes to the content of the manuscript, listed as follows:

1)             The structure-to-function relationship of lncRNAs is an essential aspect that needs to be discussed in more details. Recent studies by Lin et al (2018) doi: 10.1093/nar/gky046provided a secondary structure model for the shorter isoform hNEAT1_S indicating long range 5’-3’ interaction. Please consider discussing also the manuscript by Jiang et al., (2017) doi:10.1038/nsmb.3455

2)             Please include the discussion of NEAT1 involvement in KSHV infection and hypoxia. Refer to Coralie Viollet, David A. Davis, Shewit S. Tekeste, Martin Reczko, Joseph M. Ziegelbauer, Francesco Pezzella, Jiannis Ragoussis, Robert Yarchoan. 2017 RNA Sequencing Reveals that Kaposi Sarcoma-Associated Herpesvirus Infection Mimics Hypoxia Gene Expression Signature.

Author Response

Dear Reviewer 2

The manuscript was substantially changed and restructured according to your valuable comments and suggestions. We highly appreciate the detailed reviews which helped us improving the quality of the manuscript immensely. We tried to address every point as detailed as possible and list the responses to your comments in point-by-point fashion below.

 1. The structure-to-function relationship of lncRNAs is an essential aspect that needs to be discussed in more details. Recent studies by Lin et al (2018) doi: 10.1093/nar/gky046 provided a secondary structure model for the shorter isoform hNEAT1_S indicating long range 5’-3’ interactions. Please consider discussing also the manuscript by Jiang et al., (2017) doi: 10.1038/nsmb.3455

The structure-to-function relationship of lncRNAs and of NEAT1 as well as the suggested literature have been included in section 2.2. NEAT1 Domain Architecture.

2. Please include the discussion of NEAT1 involvement in KSHV infection and hypoxia. Refer to Coralie Viollet, David A. Davis, shewit S. Tekeste, Martin Reczko, Josoph M. Ziegelbauer, Francesco Pezzella, Jiannis Ragoussis, Robert Yarchoan. 2017 RNA Sequencing Reveals that Kaposi Sarcoma-Associated Herpesvirus Infection Mimics Hypoxia Gene Expression Signature.

We have included a discussion about NEAT1 in KSHV infection (section 3.2.2) which helped to improve the quality of the section about the pro-viral effects of NEAT1.

 Thank you very much for your efforts.

Sincerely,

Felix Prinz & Christiane Klec

Reviewer 3 Report

This is a well written and informative review on a timely topic, the function of the lncRNA NEAT1 in non-cancer diseases, that complements the previous review by the same authors on the tumorigenic role of NEAT1. Nevertheless, I find a lack of focus since this review claims to be centered in the paraspeckle-dependent mechanisms of NEAT1 function (in the Abstract section), but these are mostly absent in the text, except for the discussion on viral diseases and a short claim on their relationship to ALS development.  In this sense, this would be more a revision on NEAT1 expression in these diseases (and also a good revision) but it lacks a sound review of the paraspeckle-mediated mechanisms, not to say on A-to-I edition.

                Furthermore, the somehow artificial division of NEAT1 functions (tumorigenic vs. non-tumorigenic) in different reviews has the unwanted consequence that it also divides the mechanisms of action, paraspeckle-mediated for non-tumorigenic vs. miRNA-sponging in tumorigenesis. This makes me wonder whether the well known miRNA-sponging ability of NEAT1 is also active in non-tumor diseases. Since there are recent reports on the interaction of NEAT1 with miRNAs in non-tumor diseases (Wang et al., J Cell Physiol. 2018 Dec 13. doi: 10.1002/jcp.27959; Wang et al., J Cell Physiol. 2018 Sep 27. doi: 10.1002/jcp.27340; etc.), I think that this point should have been also treated in the review, miRNA-sponging mechanisms of NEAT1 are also active in non-tumor diseases, aren't they?.

Minor concerns.

                -Sometimes the text is very dense, and it would benefit from including more figures (e.g. the section on NEAT1 structure). In this sense Figure 1 is small and poorly informative. Perhaps this could be divided onto a figure on NEAT1 structure and another figure on NEAT1 function.

                -The manuscript should include an abbreviation list or define abbreviations the first time they appear in the text. What does NEAT1 mean? What are the IRAlu elements (line 93)?

                -Line 48 states that "NEAT1 is transcribed by Pol II from the multiple endocrine neoplasia locus in the human chromosome 11qA" . This is not fully correct and should be changed to "multiple endocrine neoplasia locus (MEN1) in the human chromosome 11q13.

                -References should be carefully checked for mistakes. Most of the citations are incomplete or include mistakes (e.g. rna instead of RNA).

 Author Response

Dear Reviewer 3

The manuscript was substantially changed and restructured according to your valuable comments and suggestions. We highly appreciate the detailed reviews which helped us improving the quality of the manuscript immensely. We tried to address every point as detailed as possible and list the responses to your comments in point-by-point fashion below.

 Nevertheless, I find a lack of focus since this review claims to be centered in the paraspeckle-dependent mechanisms of NEAT1 function (in the Abstract section), but these are mostly absent in the text, except for the discussion on viral diseases and a short claim on their relationship to ALS development. In this sense, this would be more a revision on NEAT1 expression in these diseases (and also a good revision) but it lacks a sound review of the paraspeckle-mediated mechanisms, not to say on A-to-I edition.

The whole manuscript has been restructured and we tried to highlight the characteristics and mode of NEAT1’s action in non-cancerous diseases i.e. common upregulation of NEAT1, nuclear-retention of mRNAs, transcriptional regulation due to sequestration of paraspeckle proteins and miRNA sponging. Accordingly, the abstract and conclusion sections have also been re-written. We hope that we could accentuate the essential points and focus of this review. Since nuclear-retention of A-I-edited mRNAs has only been reported in HIV-infection we mentioned this point, but did not highlight it anymore.

Furthermore, the somehow artificial division of NEAT1 functions (tumorigenic vs. non-tumorigenic) in different reviews has the unwanted consequences that it also divides the mechanism of action, paraspeckle-mediated for non-tumorigenic vs. miRNA-sponging in tumorigenesis. This makes me wonder whether the well known miRNA-sponging ability of NEAT1 is also active in non-tumor diseases. Since there are recent reports on the interaction of NEAT1 with miRNAs in non-tumor diseases (Wang et al., J Cell Physiol. 2018 Dec 13 doi: 10.1002/jcp.27959; Wang et al., J Cell Physiol. 2018 Sep 27, doi: 10.1002/jcp.27340; etc), I think that this point should have been also treated in the review, miRNA-sponging mechanism of NEAT1 are also active in non-tumor diseases, aren’t they?

miRNA sponging effects of NEAT1 in non-cancerous diseases were addressed in a paragraph in the conclusion (lines 324-339). Although it seems like NEAT1’s function in non-cancerous diseases predominantly depends on the described nuclear retention and paraspeckle-mediated sequestration mechanisms, there is evidence that NEAT1 also sponges miRNAs, thus influencing the progression of some diseases (e.g. diabetic nephropathy, atherosclerosis). Suggested publications were included in this paragraph.

Sometimes the text is very dense, and it would benefit from including more figures (e.g. the section on NEAT1 structure). In this sense Figure 1 is small and poorly informative. Perhaps this could be divided onto a figure on NEAT1 structure and another figure on NEAT1 function.

We have changed and reduced Figure 1 (now Figure 2) and think that this figure is now more informative showing the two main functions of NEAT1 within paraspeckles i.e. nuclear retention of mRNAs and sequestration of paraspeckle proteins hampering their influence on transcriptional regulation. Additionally, we have created a new figure showing NEAT1 Domain Architecture as well as its architectural conformation within paraspeckles (Figure 1) which is helping to loosen up the text.

The manuscript should include an abbreviation list or define abbreviations the first time they appear in the text. What does NEAT1 mean? What are IRAlu elements (line 93)?

According to the journal style, the abbreviations list is attached at the very end (after the reference section). The full name of NEAT1 nuclear enriched abundant transcript 1 is given in the abstract and now also at its first appearance in the main text. The full name of IRAlu elements (i.e. inverted repeated Alu elements) has been mentioned in the text and has been added to the abbreviations list.

 Line 48 states that “NEAT1 is transcribed by Pol II from the multiple endocrine neoplasia locus in the human chromosome 11qA”. This is not fully correct and should be changed to “multiple endocrine neoplasia locus (MEN1) in the human chromosome 11q13.

Thanks to the careful reading of the reviewer the correct locus of NEAT1 transcription has been included in the text.

References should be carefully checked for mistakes. Most of the citations are incomplete or include mistakes (e.g. rna instead of RNA).

We are very sorry for this mistake. The references have been changed to the predetermined guidelines and have been carefully checked for mistakes. Now the references meet the criteria of the journal and do not harbor any mistakes anymore.

 Thank you very much for your efforts.

Sincerely,

Felix Prinz & Christiane Klec
